# The Influence of Video Format on Engagement and Performance in Online Learning

**DOI:** 10.3390/brainsci11020128

**Published:** 2021-01-20

**Authors:** Sergej Lackmann, Pierre-Majorique Léger, Patrick Charland, Caroline Aubé, Jean Talbot

**Affiliations:** 1Tech3lab, HEC Montréal, Montréal, QC H3T 2A7, Canada; sergej.lackmann@hec.ca; 2Department of Teaching, UQAM, Montréal, QC H2L 2C4, Canada; charland.patrick@uqam.ca; 3Department of Management, HEC Montréal, Montréal, QC H3T 2A7, Canada; caroline.aube@hec.ca; 4Department of IT, HEC Montréal, Montréal, QC H3T 2A7, Canada; jean.talbot@hec.ca

**Keywords:** online learning, engagement, video format, learning, neuroscience

## Abstract

Millions of students follow online classes which are delivered in video format. Several studies examine the impact of these video formats on engagement and learning using explicit measures and outline the need to also investigate the implicit cognitive and emotional states of online learners. Our study compared two video formats in terms of engagement (over time) and learning in a between-subject experiment. Engagement was operationalized using explicit and implicit neurophysiological measures. Twenty-six (26) subjects participated in the study and were randomly assigned to one of two conditions based on the video shown: infographic video or lecture capture. The infographic video showed animated graphics, images, and text. The lecture capture showed a professor, providing a lecture, filmed in a classroom setting. Results suggest that lecture capture triggers greater emotional engagement over a shorter period, whereas the infographic video maintains higher emotional and cognitive engagement over longer periods of time. Regarding student learning, the infographic video contributes to significantly improved performance in matters of difficult questions. Additionally, our results suggest a significant relationship between engagement and student performance. In general, the higher the engagement, the better the student performance, although, in the case of cognitive engagement, the link is quadratic (inverted U shaped).

## 1. Introduction

Even prior to COVID-19, millions of students attended online classes that can be followed asynchronously, without real-time interaction with an instructor [1]. Many of these classes include videos that expose the learning content [2,3]. The courses and videos are developed by higher education institutions, but also by online learning platforms such as Coursera (https://www.coursera.org/). Due to differences in practices and resources available to these content creators, videos are produced in a variety of ways and their production styles vary substantially within a platform [2]. Video production style identifies the way a video’s visual aspects are organized and presented to the audience [2]. At the same time, researchers and practitioners in the field argue that choosing the right video format is crucial for student engagement and learning results, in addition to costs of production [1,3,4,5].

In traditional school settings, instructional material influences learner engagement [6]. Engagement, it is argued, lowers drop-out probability [6], and increases learning [7]. Recent studies indicate that this relationship between instructional material (video production style in particular), engagement, and student retention also holds true in the online learning environment [2,8,9]. In addition, some video production styles seem to have a larger impact on learning than others. For example, videos that include an instructor’s image result in higher student performance [10,11,12], whereas video designs that are too complex can overload a student’s cognitive capacity and decrease learning [8,12]. It is therefore necessary to analyze the impact of video lecture elements on learning performance and use those findings to improve the format [4]. Although a number of studies have been conducted that examine video production styles and their impact on engagement and learning [3], “survey data and secondary data collected via automated methods dominated the analyses” in the field [13] (p. 17). That data includes user-generated activity such as video viewing times automatically captured on online learning servers.

Furthermore, it has been demonstrated that neurophysiological measures allow us to better understand students’ experience in online learning environments [9,14]. Engagement evolves over time and is subject to important retrospective bias.

Therefore, there is a benefit to using these implicit measures to capture automatic and unconscious reactions of subjects [15,16]. Even so, no study has compared the lecture capture and the infographic video production styles by analyzing the evolution of engagement over time (the duration of the video) [17]. At the same time, there is not necessarily one optimal format [10] and not much research has been undertaken to analyze all the different formats in the field [4], such as the infographic video. According to Hansch et al. [2], we still need to understand how production value affects learning. They opine that creating the video is the most expensive aspect of producing an online course. For example, the creation of a lecture video requires content, filming equipment, a production site, and often post-editing. The process can become complex and time consuming [3,4]. Da Silva et al. [5] suggest that beyond the learning efficiency, the choice of an appropriate video design should be cost-effective. Currently, practitioners in the field justify investment in more multimedia-rich formats because these are supposed to increase learning and engagement more than less multimedia-rich material [9]. It is therefore relevant to study the gains, in terms of engagement or achievement, that are associated with these costs.

Thus, this paper aims to advance current findings by comparing the lecture capture and infographic video formats to identify the format that engages the students more emotionally and cognitively over time (the duration of the video), and that provides better learning outcomes. In addition, we further examine the relationship between student engagement and learning outcomes.

## 2. Related Work

The current literature discusses video formats using the cognitive theory of multimedia learning (CTML) [18] and the media richness theory [19]. These theoretical frameworks help to outline the conceptual differences between our two video production styles. The differences have specific consequences in terms of learning outcomes and engagement.

### 2.1. Cognitive Theory of Multimedia Learning

Mayer [18] developed the cognitive theory of multimedia learning (CTML), which aims to explain the relationship between multimedia learning and the cognitive processes of students regarding the assimilation of knowledge. This model outlines three predictions. First, a person needs to be actively processing incoming information for learning to happen. This means that a person first needs to actively use their senses (hearing, seeing) to perceive the information so that it can then be sent to and processed in the working memory [11]. Second, during “learning”, information is gathered and processed in the student’s working memory. Visual and auditory information are treated separately, in two channels [11,20,21]. The visuospatial sketchpad processes the visual information, such as an image on the screen, and the phonological loop handles auditory stimuli, such as the voice of the professor. Third, at any given time, the visuospatial sketchpad and phonological loop can handle a restricted amount of information in relation to the limited capacity of the working memory.

Regarding the first prediction, several authors in the multimedia learning field argue that active attention to the visual and verbal stimuli of the multimedia lecture is required for effective learning [11,22]. In videos, they suggest that the social cues (voice, face) of the lecturer increase the student’s engagement and therefore lead to better performance [3,12,22].

Consistent with the dual channel principle, it is suggested that video and audio input delivered at the same time provide better results in terms of engagement and learning performance in empirical studies [9]. However, these inputs in instructional videos need to be relevant in terms of improving engagement and learning, in addition to providing a positive trade-off for the cognitive load they can cause.

Too much load on the visuospatial sketchpad has been found to limit the ability of students to attend to all the content presented visually in multimedia learning, thus inhibiting the student’s acquisition of knowledge [4,12,21], and also negatively impact their engagement (attention, emotions) [4,9,12]. Further, instructor presence does not necessarily elicit better engagement, satisfaction, and perceived learning [8,12,21]. The benefits of social presence and nonverbal communication can be counterbalanced by more demand being placed on the visual channel due to the processing of social cues [11,12]. In general, the trade-offs seem to be more prominent when students are increasingly exposed to stimuli and their attentional resources are being depleted [22]. Additional research can clarify the benefits and trade-offs of different visual and audio elements.

### 2.2. Media Richness Theory

Another theoretical lens is the media richness theory. According to Trevino et al. [19], communication channels have different levels of media richness. The phrase “level of richness” denotes the extent to which a medium can efficiently transmit the information that allows people to have a prompt and common understanding. Thus, some communication means are better at transmitting information than others and allow for a rapid reduction in uncertainty and equivocality to produce the same interpretation. Four criteria allow us to determine the level of richness of a medium. First, the faster the medium allows for feedback, the better it is at reducing misunderstanding. Second, the more cues the medium provides (images, graphics, numbers, voice, body language), the better it is at facilitating the right interpretation. Third, a diverse vocabulary which includes numbers and natural language provides rigor (numbers) and context (natural language) at the same time and improves communication. Fourth, a medium which includes personal focus, in addition to emotional responses, will provide additional personal meaning and increase the efficiency of the communication [19,23].

The theoretical propositions for multimedia learning are that instructional material with higher levels of richness according to the four criteria will be better at transmitting information and conveying meaning. Empirical research suggests that rich media reduces the effort needed by parties to understand each other [4]. In multimedia learning, Chang and Chang [24], and Chen et al. [9], have demonstrated that increasing the richness of the learning material makes communication more efficient and leads to positive responses from students. Similar results were also reported by Ilioudi et al. [10], who found that lecture videos with an instructor’s image were more effective for learning complex topics than studying them from a book.

However, when Chen et al. [4] compared three different video lecture types which have seemingly different levels of richness, he found that students’ positive and negative emotional responses were similar. The research in this field can therefore benefit from further empirical studies into the effect of media richness on engagement and learning.

### 2.3. Multimedia Learning and Engagement

Many studies underline the importance of engagement in education. It improves student learning [3] and retention [25]. In particular, students’ emotions (emotional engagement) are linked to retention [17] and learning [14]. In the context of this research, it was shown that emotions (emotional engagement) can be influenced by the visual characteristics of multimedia learning material and in turn impact learning outcomes [9].

Fredricks, Blumenfeld and Paris [6] propose a multidimensional definition of engagement which is composed of behavioral engagement, emotional engagement, and cognitive engagement. Behavioral engagement is divided into three dimensions: conduct, work involvement, and participation. Conduct refers to positive behavior, such as showing up for class, or not spamming in forums. Work involvement is defined as the effort of simply doing the work. Participation means involvement in extracurricular activities, contributing to class discussions, or asking questions [6]. Given the passive research’s context (i.e., online videos without behavioral interaction with instructors and peers), we have excluded this dimension from our study to focus on emotional and cognitive engagement.

Emotional engagement is broadly defined as positive or negative reactions that do not require cognitive effort and that are spontaneous [16]. Such reactions can be discrete emotions such as interest, boredom, happiness, sadness, and anxiety [6]. In turn, these discrete emotions can be grouped under arousal (i.e., activation) and valence [14]. Feldman [26] defines valence as a pleasant or unpleasant emotional state (e.g., happiness, sadness), whereas arousal describes a physiologically aroused or calm state (e.g., anxiety, boredom).

A person is cognitively engaged when actively trying to understand new information [6]. In particular, cognitive engagement is a “psychological process involving attention and investment” [27] (p. 18). Fredricks, Blumenfeld and Paris [6] support the hypothesis that it is necessary to study the three engagement components together, because they are interlinked and together provide a richer characterization of a student’s state.

## 3. Hypothesis Development

As stated previously, our general research objective is to analyze two different video formats (lecture capture and infographic video) in terms of engagement (RQ1) and learning (RQ2). These formats are characterized by social cues, cognitive demand, and media richness. As supported by our literature review, these characteristics have an impact on the emotional and cognitive engagement of students. Additionally, the aim of this paper is to further examine the link between engagement and learning (RQ3).

We argue that our two video production styles are characterized by a different level of media richness, and that this difference in media richness will produce a significant disparity in emotional engagement. In this regard, the infographic video provides more visual cues (second criteria) and richer vocabulary (third criteria) than the lecture capture, which has only simple video and audio. This will produce more excitement (arousal) and satisfaction (valence) in the infographic condition. Furthermore, human presence in the lecture capture can activate substantial cognitive resources, increase the effort required to process the information and offset the benefits provided by social cueing [4,12]. In light of the CTML and the media richness theory, we propose that emotional engagement (arousal, valence) will be higher in the infographic condition.

**Hypothesis** **1A.**
*Emotional engagement will be higher for infographic video.*


As discussed for emotional engagement, we argue that the rich content of the infographic video sustains a student’s attentional engagement more effectively than a mere focus on the teacher for 15 min, similar to the results of Korving et al.’s [22] research. Maintaining focus on a single static point can be challenging. Thus, dynamic content with short shot sequences can be better at maintaining a student’s cognitive attention [5].

Chen et al.’s [9] results also suggest that richer media provides a more immersive experience for students. Objects which appear on the screen such as tables and images are well-organized in the infographic condition. There is no second window which shows the professor or agenda. Therefore, the learner can focus all his attention on the information displayed without having to divide it between the professor’s image and the text, for example. In conclusion, we argue that cognitive engagement will be higher in the infographic condition.

**Hypothesis** **1B.**
*Cognitive engagement will be higher for infographic video.*


Most research supports the hypothesis that higher media richness has a positive impact on learning performance, as in the case of infographic video compared to lecture capture. Interesting visual effects and graphics which convey further meaning and provide more context are likely to support more learning, especially in terms of difficult questions. In addition, according to the CTML, multimedia material which provides audio and video stimuli will promote better learning. An infographic video presents visual information in the form of graphics which are processed by the visual loop. At the same time, the narration provides additional explanations which are processed by the audio loop of the learner. By comparison, the lecture capture provides narration and an image of a teacher in the class. However, a simple image of the teacher does not necessarily convey or support additional information processing in the video loop [21]. Therefore, we propose that the infographic video format will have a better impact on learning.

**Hypothesis** **2.**
*Learning performance will be higher for infographic video.*


As discussed, many studies argue that there is a significant relationship between the emotional engagement of students and their learning [8,9,14,21]. In particular, Lee et al.’s [8] research shows that socialness perception, arousal, and pleasure have an effect on a student’s performance. While testing different video course designs, Chen et al. [9] found that there is a link between negative emotions and students’ lower learning. We expect a positive correlation between positive emotional states and performance, so we propose that the higher the emotional engagement, the better the student performance.

**Hypothesis** **3A.**
*The higher the emotional engagement, the better the student performance.*


Cognitive engagement and, more specifically, attention, is considered key for effective learning in multimedia teaching [22,28]. Chen et al. [4] further proposes that video production style is an important focus of analysis because it could affect sustained attention. He argues that excessively high levels (and deviation) of sustained attention cause stress and, as a result, students’ performance decreases. Thus, we expect a positive correlation between attention and performance. However, this relationship is likely to be quadratic (inverted U shaped).

**Hypothesis** **3B.**
*There is a quadratic relationship between cognitive engagement and performance.*


## 4. Methods

### 4.1. Experimental Design and Participants

To test these hypotheses, we conducted a between-subject lab experiment with two conditions. In both conditions the subjects passively watched an approximately 14-min-long pedagogical video on a segment of a course on Introduction to Organizational Behavior. During viewing, emotional and cognitive measures were recorded with neurophysiological instruments (Noldus Information Technology, Wageningen, Netherlands; Biopac, Goleta, USA; Brain Vision, Morrisville, USA; Mensia Technologies, Paris, France). Furthermore, pre- and post-test multiple-choice questionnaires were administered to measure student learning. Upon completion of the experiment, subjects were given a $30 gift card for the university bookstore. The research protocol was approved by the HEC Montréal-Comité d’éthique de la recherche (2017-2364-55) before the experiment took place. A total of 16 male and 10 female subjects participated in the study. They were recruited using an institutional panel of research participants. The subjects were on average 26.20 years old with a standard deviation of 6.93. Male participants had an average age of 25.56 (standard deviation (STD) 6.93), and female 27.33 (STD 6.98).

Subjects were randomly assigned to one of the two conditions and were pre-screened before participation. In the infographic condition, the mean age was 24.33 (STD 6.93). There were nine males (mean age 22.22, STD 6.93) and three females (mean age 30.67, STD 7.58). In the lecture condition. the average age was 27.92 (STD 7.11), with seven males (mean age 29.86, STD 7.17) and seven females (mean age 25.67, STD 6.98). First, only subjects who had not previously attended any psychology classes at the university level were admitted. This was done to ensure that no one had prior knowledge of the content in the videos. Second, subjects who had a psychological diagnosis such as epilepsy were excluded as this could cause deviations in the electroencephalography (EEG) signal, similar to Gollan et al. [29].

### 4.2. Experimental Stimuli

Two lecture videos with the assistance of the same professor were created specifically for the study. Both videos explained exactly the same content, namely, a chapter of the introductory psychology course at our institution, and had approximately the same length (14 min). This is within the length range suggested for videos by previous research [4,22,30]. In the rich infographic condition, learners saw a continuous flow of images, graphics, and text which was synchronized with an audio track (infographic video, see Figure 1); this condition was designed building upon the best practices suggested by the edX community on how to build and run an online course (visit edx.readthedocs.io for more details) In the lecture condition, subjects saw a video recording of a class lecture (lecture capture, see Figure 2) that was not enriched. A professor and some students who attended the class are visible. No other content, such as PowerPoint slides, was present. The voice of the professor was the same in both conditions.

Like in Chen et al. [4], Table 1 summarizes the characteristics of our video production styles in terms of the theory discussed (media richness and CTML).

### 4.3. Operationalization of the Research Variables

The students’ emotional and cognitive engagement over time was inferred on the basis of the related neurophysiological states (valence, arousal, attention) which we measured using implicit and explicit data. Implicit data was collected through neurophysiological sensors (e.g., electroencephalography (EEG), electrodermal activity (EDA), and facial expression recognition). Explicit data was self-assessed (e.g., questionnaires). Table 2 illustrates the operationalization.

#### 4.3.1. Emotional Engagement

First, it needs to be noted that emotions are considered multi-componential, and they can cause different bodily responses (e.g., open mouth, increased heart rate) [14]. These emotional components are divided into behavioral (facial), experimental (how emotions make one feel), and physiological (EDA) responses [14]. In other words, “emotional response can be measured in at least three different systems-affective reports, physiological reactivity, and overt behavioral acts” [14] (p. 49). Affective reports (such as the Self-Assessment Manikin (SAM) Scale) help us understand how participants perceive and feel about the stimuli [31].

In particular, the SAM scale measures the self-reported affective state (i.e., pleasure and arousal) using a non-verbal scale [31]. Because the SAM scale is iconographic and contains only a single item, research shows that it seems to capture a more spontaneous emotional perception [32]. The SAM scale is a popular instrument and has been previously used in many fields, including education research. For example, Lee et al. [8] used the SAM to compare different video production styles. Consequently, we used the SAM to capture the self-reported arousal and valence of participants. After the learners viewed the video, a web page featuring the SAM appeared on the screen.

In addition, Keltner and Ekman [33] outlined the link between facial expressions and six basic emotions (happiness, sadness, surprise, fear, disgust, and anger). Based on that work, several studies were able to successfully use automatic facial expressions analysis software to recognize emotions of participants in an e-learning environment [34,35,36]. In this study, Noldus FaceReader (Noldus Information Technology, Wageningen, Netherlands) was used as automatic facial emotion recognition software. It records the subject’s facial expressions and measures the intensity of Ekman’s six basic emotions and the neutral expression of the participants. To be specific, the Active Appearance Model captures the facial expressions, while an artificial neural network classifies them and computes a standardized valence and arousal value. The value reported ranges from −1 to 1 with 30 inferences per second. The higher the value, the more the person is pleased or aroused. FaceReader has been validated and used in studies similar to ours [14,37].

Physiological reactivity in the form of EDA can be measured by looking at the skin’s electrical conductance levels (SCL) and changes in these levels due to sympathetic activity. For that purpose, two electrodes pass a small amount of electricity through the skin and SCL is measured in microsiemens or µS. EDA (arousal) is high when people are curious or anxious, and low then they are bored/relaxed [14]. Electrodermal activity is widely and reliably used to measure arousal due to stimuli [16]. In particular, Charland et al. [38] and Harley et al. [14] used EDA to assess students’ arousal states during learning exercises. We used Biopac’s technology (Biopac, Goleta, CA, USA) to record EDA data. To capture electrodermal activity, a wireless amplifier Biopac MP (Biopac, Goleta, CA, USA) and two electrodes placed on the palm of the non-dominant hand were used as apparatus [39]. Biopac MP has been successfully employed in a variety of studies to measure arousal [39,40].

#### 4.3.2. Cognitive Engagement

Based on the frequency and amplitude of the signal, and the spatial location of its origin, EEG can be used to infer cognitive states of the subjects. In particular, it allows us to assess attention [28,41]. Chen et al. [4] used an EEG headset to measure sustained attention. He was able to detect low attention spans in students who were watching video lectures.

Pope, Bogart and Bartolome [42] developed a cognitive engagement index (vigilance index) based on the EEG signal’s spectral decomposition. This engagement index has been widely used in different contexts [38]. They mainly argue that higher beta activity is related to increased vigilance, whereas increased alpha and theta activity is linked to lower vigilance. Subjects in a state of high vigilance (high beta activity) respond better to stimuli [41].

We computed our cognitive engagement ratio according to a modified version of the cognitive engagement index as proposed by Mikulka et al. [43] using Mensia NeuroRT (Mensia Technologies, Paris, France), advanced signal processing software. Building upon previous work [42], a cognitive engagement index was designed to range from 2 to 20. Our EEG data collection was performed with BrainVision ActiChamp 32 (Brain Vision, Morrisville, NY, USA), which is a non-invasive technique that uses electrodes placed on the scalp surface to record EEG signals [9]. We used the international 10/20 system for electrode placement as proposed by the American Encephalographic Society [44].

#### 4.3.3. Learning Performance

To measure learning performance (achievement), subjects completed a multiple-choice questionnaire (25 questions), before (pre-test) and after (post-test) viewing either version of the learning video. Similar to previous research, there was no time limit to complete the questionnaire, and no feedback was given to avoid affecting the post-test [4,16,21]. Pre-test and post-test questions are the same, as in Chen et al. [9]. Recall and transfer questions were included, as used in Wang et al. [12]. Each question was answered either correctly and counted as 1, or incorrectly and counted as 0. For each student, the percentage of correctly answered questions was calculated for the overall questionnaire and for each difficulty rating. Learning was calculated by subtracting pre- and post-test results.

To create the difficulty classification, the complexity of questions was rated by six experts using a 4-point Likert scale (1 = easy, 2 = medium, 3 = medium-difficult, 4 = difficult), as suggested by Cronan et al. [45] and similar to previous research [12]. Questions that were answered incorrectly by ≥50% of the experts were removed as ambiguous.

The median difficulty of all questions based on an expert rating was 2. Easy questions measure low complexity knowledge, whereas difficult questions measure high complexity knowledge. According to Bloom’s revised taxonomy, students who can answer actually difficult questions have acquired a deeper understanding of the material [45]. They can analyze a new problem using acquired knowledge and apply that knowledge to solve the problem.

### 4.4. Experimental Setup and Protocol

The experiment was conducted in a usability laboratory. Participants were seated in an experimental room with a computer to perform the task. In an adjacent room, the researchers monitored the experiment via a one-way mirror and computers to record the participants’ neurophysiological responses.

There were no breaks during the experiment, and each session lasted on average 1 h and 30 min, including the setup and calibration of the equipment. Upon arrival, subjects had to first read and sign the Consent Agreement and their conformity to our selection criteria was verified. Next, they were informed that they would be taking part in a study which evaluates different lecture video designs, and they were given a summary of the experiment.

Once the equipment was set up, and the EEG baseline was established, and all of the devices began their recordings, which continued until all of the tasks in the experiment were completed. The data was synchronized using markers which delineated each part of the experiment in all of the equipment (e.g., beginning/end of the video). All tasks were performed on a 22-inch flat screen in front of the participants. Subjects were first asked to relax and close their eyes for 90 s to provide a baseline for the EEG signal. Then, for one minute, participants saw and counted randomly colored squares which appeared on the screen for 6 s each. In the next step, they completed a multiple-choice questionnaire (25 questions) to assess their pre-test knowledge of the video content. After completing the assessment, subjects started to watch one of the two videos.

No note-taking or pausing was allowed during viewing to ensure comparability of the neurophysiological data over time between the two conditions and subjects. Once the video ended, a SAM questionnaire was administered. As a post-test, the same questions as in the pre-test were asked, to assess the learner’s performance after viewing the video.

### 4.5. Data Processing

Once acquired, all of the data—i.e., EDA data from Biopac, FaceReader’s valence and arousal, cognitive engagement ratio from Mensia NeuroRT, SAM results, and pre- and post-test answers—was exported into csv files and imported to SAS (SAS Institute Inc., Cary, NC, USA). Data was then prepared, synchronized, and analyzed in SAS using statistical models appropriate for each research question/data type.

For implicit granular-over-time data, we first computed a one second moving average for valence and arousal similar to previous research [14,37]. Given the experiment’s 14 min recording, we obtained 840 one-second epochs (60 s × 14 min) for each participant. Measures that had distributions that departed greatly from a normal distribution were transformed using log or square root transformations [46]. A one second average of the cognitive engagement index was calculated using Mensia NeuroRT. Then, we used repeated-measures multiple linear regression with random effects (subject) to test if there is a significant difference between conditions (over time/on average) in terms of arousal, valence, or attention, similar to Sanders [47] and Charland et al. [41]. We argued that neurophysiological data can be modeled using time series because fluctuations depend on past values. Thus, we used “proc mixed” with autoregressive covariance structure (see SAS/STAT(R) 9.2 User’s Guide, Second Edition). Furthermore, we included random effects for the origin/intercept to account for variability between individuals. The method used was maximum likelihood. Additionally, we used the Mann–Whitney U Test because of the small sample to compare the SAM’s valence/arousal between the two conditions [10]. Moreover, we computed the subjects’ performance for the pre- and post-test questionnaires and calculated how much they had learned. To verify if there were significant differences in learning between conditions, we performed a signed rank test.

Finally, we used multiple linear regression (“proc reg”) to model the relationship between the performance as dependent variable and a given implicit neurophysiological measure as independent variable. The overall implicit measures are averages calculated over the video session (i.e., the average of each of the 840 one-second data points recorded for each participant). We also included quadratic relationships and condition as a control variable in the model. The relationship between the SAM’s valence/arousal and performance was analyzed using Spearman correlation coefficients due to the small sample size.

## 5. Results and Discussion

We compared lecture capture and infographic video formats by considering engagement over time (RQ1) and student performance (RQ2). An analysis of the link between student engagement and learning outcomes was also performed (RQ3). For RQ1, we hypothesized that emotional (H1A) and cognitive (H1B) engagement are higher for the infographic video. The latter should also allow for better learning outcomes (RQ2-H2). Finally, for RQ3, we proposed that the higher the emotional (H3A) and cognitive (H3B) engagement, the better the student performance. This relationship is, however, quadratic in the case of cognitive engagement.

### 5.1. Comparison of Emotional Engagement between the Conditions (H1A)

The results summarized in Table 3 suggest that, on average, there was no significant difference in valence between the conditions (*β* = −0.19, Sig. = 0.15 > 0.05) according to facial expression analysis. However, there was a significant difference in the evolution of valence over time (*β* = 7.30 × 10^−6^, Sig.≤ 0.0001 < 0.05). Although valence of students who viewed the infographic video decreased over time, it increased over time for those who viewed the lecture capture.

Explicit valence as reported with the SAM seems to confirm the above finding, as there was no significant difference (on average) in self-reported valence between the conditions (Sig. = 0.24 > 0.05). In general, subjects in both conditions were, however, pleased with the videos: infographic video (mean 6.17, STD 1.47) and lecture capture (mean 6.86, STD 1.66).

Furthermore, there was no significant difference in arousal between the conditions (*β* = −0.02, Sig. = 0.26 > 0.05) according to facial expression analysis. On average, the subjects in both infographic and lecture conditions had similar arousals. However, there was a significant difference in the evolution of arousal over time (*β* = −5 × 10^−6^, Sig. ≤ 0.0001 < 0.05). Although arousal of students who viewed the lecture capture decreased over time, it increased over time for those who viewed the infographic video.

Self-reported arousal seems to confirm the above finding, as there was no significant difference (on average) between the conditions (Sig. = 0.53 > 0.05). In general, subjects in both conditions were somewhat excited while watching the infographic video (mean 5.42, STD 1.62) and the lecture capture (mean 4.79, STD 2.39).

Regarding arousal operationalized through mean EDA, there was a significant difference between the two conditions (*β* = 0.24, Sig. ≤ 0.0001 < 0.05). Subjects in the infographic condition had, on average, lower arousal than subjects in the lecture condition. However, over time, the infographic video was able to incite significantly more arousal than the lecture capture (*β* = −60 × 10^−6^, Sig. ≤ 0.0001 < 0.05). At some point, around 7 min, the former managed to invoke more arousal than the latter.

It is relevant to note that arousal in both conditions decreased over time. This was not observed in the case of implicit arousal deduced from facial expressions. Furthermore, the latter and explicit arousal did not exhibit a significant difference between conditions on average. Nevertheless, both implicit measures (facial emotions and mean EDA) show that lecture capture induced less arousal over time than infographic video.

Regarding deviation in implicit arousal according to EDA recordings, our results suggest that there was no significant difference between the conditions on average (*β* = −0.004, Sig. = 0.58 > 0.05). Subjects who watched the infographic video had, on average, a similar arousal variability to those who watched the lecture capture. However, over time, the infographic video was able to incite significantly more arousal deviation than the lecture capture (*β* = 2.4 × 10^−6^, Sig. = 0.02 < 0.05). In addition, after 3 min, the former managed to produce more arousal variability than the latter. This result was supported by implicit arousal operationalized using facial expressions and mean EDA. It is interesting to note that variability of EDA in both conditions decreased. We can assume that students get tired over time and their EDA variability stabilizes as their arousal response to external stimuli diminishes.

The results do not support H1A: Emotional engagement will be higher for the infographic video. The infographic video could maintain the arousal of subjects over time significantly better than can lecture capture. However, the lecture capture video format seemed to engage subjects more emotionally at the beginning of the viewing period and students were happier to watch it over time. Specifically, we argue that subjects seemed to be more aroused (higher mean EDA) when seeing the professor at the beginning of the video and responded positively (higher valence) over time to the social presence provided by the lecturer.

From the theoretical standpoint, the CTML framework suggests that the lecturer in the video will have a positive effect on students’ emotional engagement by providing social cues. Regarding the media richness theory, the lecture capture shows the professors’ body movements, which could provide additional cues, increase personal meaning (fourth criteria), and affect arousal and valence positively. In empirical studies, it has been shown that social perception is related to arousal [8] and that viewing the lecturer increases student learning, satisfaction, and (affective) engagement [11,12]. In particular, when comparing lectures that include PowerPoint slides and/or no professor visuals, Kizilcec et al. [11] report that students prefer lectures in which the professor is shown. In our study, we saw a spike in mean EDA at the beginning for the lecture condition because the human face provides an “intimate and personal” [3] (p. 5) feel compared to PowerPoint slides.

Nevertheless, the infographic video is better at maintaining arousal. It provides more visual and vocal cues. In coherence with the media richness theory and related studies [9], this leads over time to higher emotional engagement, in our case arousal, compared to lecture capture, which is not enriched. At the same time, the social cues of the professor create an additional strain on the students over time as suggested by previous research [4,12]. Thus, emotional activation decreases faster in the lecture condition.

### 5.2. Comparison of Cognitive Engagement between the Conditions (H1B)

Our results suggest that on average, there was no significant difference in attention between the conditions (*β* = −0.06, Sig. = 0.81 > 0.05). On average, the subjects in both infographic and lecture conditions had similar attention levels. However, there was a significant difference in the evolution of attention over time (*β* = −44 × 10^−6^, Sig. = 0.03 < 0.05). While the attention of subjects in infographic condition increased, it decreased for those in lecture condition. In other words, although attention in both conditions began similarly (significant intercept), over time the infographic video engaged the students much more. Furthermore, after 2 min the latter managed to evoke more attention than the lecture capture. Therefore, the results do support H1B: Cognitive engagement will be higher for the infographic video.

Previous research supports the hypothesis that richer media will provide a more immersive experience to learners than less rich media, in our case, lecture capture [9]. In addition, based on cinemetrics analysis, Da Silva et al. [5] recommend that online learning videos should be dynamic and have short shot sequences to attract a student’s attention. The infographic video has interesting visual effects, animated graphics, text, and audio. Thus, we argue that the higher media richness of the infographic video allows for better cognitive engagement over time, compared to not-enriched lecture capture.

Furthermore, the two channels hypothesis of the CTML suggests that students will perform better if the multimedia learning material provides both visual and auditory stimulus [5]. For instance, an instructor’s face in addition to his speech contributes supplementary social cues which can evoke more attention and emotion.

However, as in the case of the lecture capture, such benefits can be offset with a greater cognitive load, burden the students, and negatively impact engagement and learning [4,9,12]. This could especially be the situation if the instructor’s image does not provide additional information to that already being presented on the slides or through narration [21]. Similar to our results, the first outcome of Korving et al.’s [22] research suggests that subjects pay less attention when the professor is visible, than when he is not. Consequently, as the infographic video integrates visual and auditory stimuli more effectively, it is better at preserving subject attention over time than the lecture capture.

### 5.3. Comparison of Learning Outcomes Between the Conditions (H2)

Before seeing the videos, students answered 53% of questions correctly overall. The percentage was 51% in the infographic condition and 54% in the lecture condition. There was no significant difference in the overall performance between the two conditions pre-test (Sig. = 0.68 > 0.05). In addition, there was no difference between the two groups in mastering the different types of questions (easy: Sig. = 0.79 > 0.05, medium: Sig. = 0.75 > 0.05 difficult: Sig. = 0.11 > 0.05). The post-test questionnaire revealed that students improved significantly after watching the video. Students performed significantly better after watching the videos, regardless of the condition and question type (Sig. = 0.04 < 0.05).

In post-test questionnaires, participants in the infographic condition performed 3% better overall and their performance increased by 6% more. However, this overall difference between conditions was not significant (Sig. = 0.20 > 0.05). Examination of the results of the different question types separately shows that there was a significant difference in terms of mastering difficult questions (Sig. = 0.04 < 0.05). The number of correctly answered difficult questions increased by 38% in the infographic condition and only by 18% in the lecture condition. Four of six difficult questions were answered better by subjects in the infographic condition. The results support H2: Learning performance will be higher for infographic video. It is relevant to note that a significant difference for all question types would have been problematic. Both videos exhibit the same content. Therefore, there should not be a significant difference between the two conditions in terms of answering easy questions. The opposite could have indicated a design issue with our videos.

The findings are supported by the media richness theory. Because the infographic video is a richer medium than the lecture capture, it is better at reducing uncertainty and equivocality. Thus, students are able to learn faster and better. Empirically, our results are supported by earlier research asserting that richer media improves student performance [4,9,10]. As is the case with the infographic video, Chen et al. [4] outline in their study that some formats have an improved layout, with better presentation of verbal and nonverbal elements than others, and therefore are more efficient at teaching students. Likewise, Illioudi et al. [10] report that richer media specifically allows students to master difficult topics.

According to the CTML, enhanced integration of video and audio allows improved learning because information is processed by two separate channels [4]. In this regard, several studies have indicated that a professor’s image adds social cues and meaning (gesturing), and therefore increases learning [10,21]. When students see a professor, social interaction schemas are activated, which aids cognitive processing. At the same time, students can focus their visual attention on the lecturer and listen to audio simultaneously. This leads to improved comprehension because information streams from two channels can complement and enhance each other [12]. However, our results indicate that the aforementioned does not necessarily translate into more learning with the lecture capture. Conversely, Wang et al. [12] and Homer et al. [21] outline in their research that the instructor’s image does not necessarily provide much more information, and social cues can be offset by the increased cognitive load they cause.

### 5.4. Relationship between Emotional Engagement and Learning (H3A)

The results, summarized in Table 4, support the hypothesis that there is a significant relationship between a subject’s overall performance and valence, based on facial expression analysis. The higher the valence, the higher the performance (adjusted *R* = 0.13, *β* = 20.27, Sig. = 0.03 < 0.05). However, self-reported valence seems to contradict the above finding, because there was no significant correlation between explicit valence and learning outcomes (Sig. = 0.53 > 0.05). Furthermore, there was a significant relationship between a subject’s overall performance and mean arousal (facial expression analysis), if we include deviation of arousal into the model. The higher the mean and variability of arousal, the higher the performance (adjusted *R* = 0.11, *β* = 123.62, Sig. = 0.03 < 0.05 (one-tailed) and *β* = 110.22, Sig. = 0.04 < 0.05 (one-tailed)). This effect was true regardless of the condition, as can be seen in Table 4.

According to EDA measurements, there was also a significant relationship between mean arousal and learning outcome, when we include deviation of arousal into the model. In particular, the higher the mean arousal, the higher the performance (*β* = 110.22, Sig. = 0.04 < 0.05). As deviation of arousal increased, performance decreased (*β* = −16.19, Sig. = 0.02 < 0.05). This effect was true regardless of the condition. It is important to note that this significant relationship was only observable if we calculated mean EDA for the first 5 min of the viewing time. There was no significant relationship between mean EDA and performance if we took the average of the whole video session. This result can be explained by the evolution of arousal over time (H1A). Over the duration of the video, mean EDA decreased significantly for both conditions (*β* = −13 × 10^−6^, Sig. = 0.005 < 0.05). As subjects became tired, mean EDA was lower at the end of the video viewing period for both conditions, regardless of the performance of the students. This masks the relationship between high mean EDA and high performance.

Explicit arousal reported through the SAM partially confirms the above results. There was no significant correlation between overall self-reported arousal and learning outcome. However, if we analyze the conditions separately, arousal in the infographic condition had a significant positive correlation (Sig. = 0.03 < 0.05) with performance.

In summary, our results suggest a significant positive relationship between emotional engagement and performance, in support of H3A: The higher the emotional engagement, the better the student performance. A decrease in negative emotions and a pleasant emotional state (valence) allowed subjects to focus on the learning task and perform better. This result is widely supported by current literature [8,9,14,21].

The effect of variability of arousal warrants further investigation because results are inconsistent between EDA and facial expression analysis. Both measures could represent a different element of arousal and therefore show contrasting effects on performance [37]. When Harley et al. [14] compared facial expression analysis results, questionnaire reports, and EDA data in an e-learning study, they could not find a clear relationship between the measures.

### 5.5. Relationship between Cognitive Engagement and Learning (H3B)

The results suggest a quadratic relationship between attention and performance (Sig. = 0.04 < 0.05, adjusted *R* = 0.07). First, the higher the attention, the better the performance. However, at some point, attention has a negative effect on performance; the average attention varied between 4.29 and 7.18 and performance appeared to peak around 5.53. Too much attention can cause a decrease in performance as the students’ attentional resources get overloaded. This effect was found in both conditions. Previous research outlines that attention is a pre-requisite for processing information and learning [22]. Furthermore, a related study specifies that attention is correlated with learning outcomes. Moments during the video lecture where students demonstrated low attention had a negative correlation with post-test scores [9]. In addition, our results align with those of Chen et al. [4]. When too much attention is required, student performance decreases. Chen et al. [4] compared a classroom lecture, a voice-over presentation, and a picture-in-picture video lecture. The voice-over presentation has the highest mean and deviation of attention. At the same time, this type of video lecture has the lowest learning performance compared to the other two. Chen et al. [4] argue that there is a split attention effect as students have to focus on the teacher visuals, slides, and table of contents at the same time. That makes it more difficult to pay attention and process information.

It is important to note that the significant relationship was only observable if we calculated the mean attention for the initial two-thirds of the viewing time (500 s). There was no significant relationship between attention and performance if we took the average of the whole video session. This can be explained by the evolution of attention (H1A). Over time attention significantly decreased in both conditions (*β* = −38 × 10^−6^, Sig. = 0.01 < 0.05). As subjects became tired, mean attention was lower at the end of the viewing period, regardless of the performance of the students. This masks the relationship between high attention and high performance. The same circumstances have been observed for arousal measured by EDA, as discussed in Hetland et al.’s [37] article.

Finally, as per regression analysis, engagement does not mediate the relationship between condition (video format) and students’ achievement. This outcome is expected. According to our results, engagement influenced overall performance (H3A/B). At the same time, if we compare the conditions, a student’s performance was only different in regard to difficult questions (H2). There was no effect of the condition on students’ overall performance. Consequently, there can be no relationship between condition and overall performance that is mediated by engagement.

### 5.6. Limitations

As with any experiment, this research has limitations. Participants were only passively attending a single video. In a typical course setting, learners would be watching a sequence of online videos over a specific duration over time. Our experiment setup could not capture the effect of time, and other research methods will be needed to further investigate the longitudinal impact of video format. In addition, we only analyzed two different video production styles. There are obviously many video formats that could be compared using a similar methodology, including hybrid options using audio-visual aids in the lecture capture.

Furthermore, we did not manipulate the quality of the rich media condition. Although we followed edX guidelines to develop the infographic condition, we acknowledge that various levels of richness quality might have a moderating effect on the learner’s engagement, and future research is needed to explore this effect.

Additionally, the length of our videos was 14 min. It would be pertinent to explore if our results hold over shorter or longer duration of videos, and examine how performance and engagement vary throughout an entire online course.

Finally, although our sample size is common for neurophysiological study, and our research provides a unique and rich perspective on the learner’s emotional and cognitive response, our findings should be replicated with a larger sample and a longitudinal research design to determine the extent to which this effect is observed over time.

## 6. Conclusions

Researchers in the field suggest the most suitable video format should be selected based on how it affects the performance and engagement of students [4]. To advance the current literature and assist with this choice, our objective was to compare two currently used video lecture formats: infographic video and lecture capture. The measures of comparison were student performance and engagement. In addition, we analyzed the relationship between engagement and learning outcomes.

As can be seen in Table 5, our hypothesis (H1A) on emotional engagement was not supported. On average, arousal (EDA) was higher for lecture capture and over time students seemed to be happier (valence) with that format. However, the infographic video was better at keeping students aroused over time.

Regarding cognitive engagement, our hypothesis (H1B) was supported. The infographic video retained students’ attention significantly better over time. In addition, students significantly improved in terms of correctly answered difficult questions in the infographic condition (H2). Finally, there was a noticeable relationship between learning outcomes and engagement (H3A/B). In general, the results suggest that the higher the engagement, the higher the performance. Too much attention, however, decreases performance at some point when the students become overwhelmed with information.

Considering these results, it seems that students perform better in the infographic condition because the format is better at keeping students aroused and attentive over time. Even though subjects seem to be specifically aroused (EDA) at the beginning upon seeing the professor and are happier over time, the lecture capture is unable to maintain higher levels of attention and arousal. Furthermore, happiness does not seem to be enough to offset the decrease in arousal and attention over time and to encourage as much learning as the infographic video. Consequently, a combination of the two formats (rich media content with social cues) could yield better engagement and learning. Rich content would help maintain arousal and attention over time, and timely social cues would bring in additional excitement and increase happiness of students.

Future research could benefit from incorporating neuropsychological measures into the study design to gain better insight into students’ engagement, and to capture automatic, in addition to unconscious, reactions of subjects over time. Researchers could also perform a more granular analysis of neurophysiological data and try to understand how specific video design components affect engagement. Furthermore, although subjects in this study were mostly young students and we had a relatively small sample size, the online learning population is highly diverse and large. In the future, researchers could examine how different subpopulations engage with learning content using a larger number of subjects. Lastly, subjects were monitored during the whole experiment which could force engagement, and no breaks or note-taking were allowed. In real life, students mostly follow online courses at home, and subsequent studies could better mimic those conditions while capturing neurophysiological data.

## Figures and Tables

**Figure 1 brainsci-11-00128-f001:**
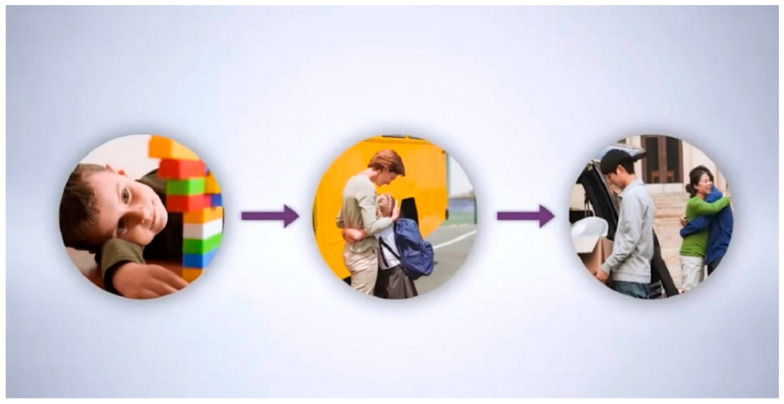
Infographic condition screenshot.

**Figure 2 brainsci-11-00128-f002:**
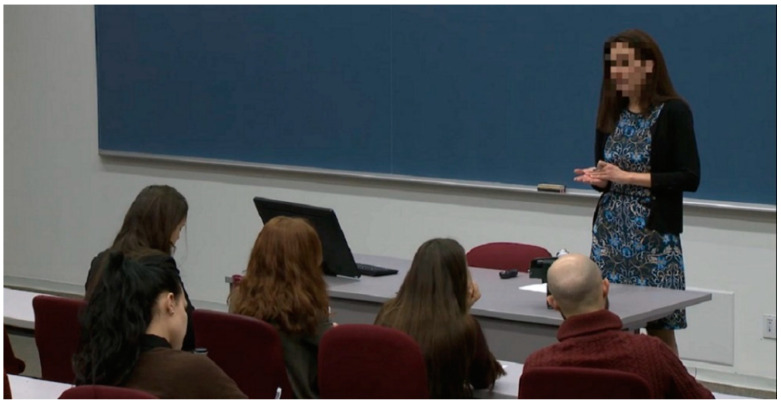
Lecture condition screenshot.

**Table 1 brainsci-11-00128-t001:** Research stimuli in comparison.

Comparison Item	Lecture Capture	Infographic Video
Cost	Low	High
Conveyed learning context	A professor presenting the subject in a traditional class setting	A visual presentation which shows dynamic content with a background voice of the same professor
Multimedia elements	Camera focused on the professor, audio	Graphics, images, text, audio
Media richness	Medium	High
Social cues	Many	Some

**Table 2 brainsci-11-00128-t002:** Operationalization of the measures.

Measure	Neurophysiological State (Response to Stimuli)	Operationalization
Emotional engagement	Affective response: valence	Implicit measure: Facial expressions (Noldus Information Technology, Wageningen, Netherlands) Explicit measure: SAM Scale-pleasure
Emotional engagement	Affective response: arousal	Implicit measures: Facial expressions (Noldus Information Technology, Wageningen, Netherlands) and EDA (Biopac, Goleta, USA) Explicit measure: SAM Scale-arousal
Cognitive engagement	Cognitive response: attention	Implicit measure: EEG (Brain Vision, Morrisville, USA), Mensia NeuroRT (Mensia Technologies, Paris, France)
Learning performance		Difference between pre- and post-test multiple-choice questionnaire results

Self-Assessment Manikin (SAM) Scale, see [31].

**Table 3 brainsci-11-00128-t003:** Engagement results for lecture condition compared to infographic condition.

	Facial Expressions				SAM		EDA				EEG	
	Valence (Mean)	Valence (Over Time)	Arousal (Mean)	Arousal (Over Time)	Valence	Arousal	Arousal (Mean)	Arousal (Over Time)	Arousal (STD)	Arousal (STD Over Time)	Attention (Mean)	Attention (Over Time)
*β* Estimate	−0.19	7.30 × 10^−6^	−0.02	−5 × 10^−6^			0.24	−60 × 10^−6^	−0.004	2.4 × 10^−6^	−0.06	−44 × 10^−6^
*t* Value	−1.45	4.97	−1.12	−10.73			8.22	−9.48	−0.55	2.43	−0.23	−2.11
Sig.	0.15	<0.0001	0.26	<0.0001	0.24	0.53	<0.0001	<0.0001	0.58	0.02	0.81	0.03

Note: Infographic condition is used as reference.

**Table 4 brainsci-11-00128-t004:** Regression results for the relationship between engagement measures and learning outcomes.

	Facial Expressions					SAM		EDA			EEG		
	Intercept	Valance (Mean)	Intercept	Arousal (Mean)	Arousal (STD)	Valance	Arousal (Infographic Video)	Intercept	EDA (Mean)	EDA (STD)	Intercept	Attention (Mean)	Attention (Squared)
*β* Estimate	77.17	20.27	13.92	123.62	110.22			69.37	9.30	−16.19	−680.62	889.34	−259.85
*t* Value	17.97	2.25	0.52	2.01	1.83			10.01	2.14	−2.38	−1.87	2.09	−2.11
Sig.	<0.0001	0.03	0.61	0.06	0.08	0.53	0.03	<0.0001	0.04	0.02	0.08	0.04	0.04

Note: Condition is included in all regressions as a control variable and is not significant.

**Table 5 brainsci-11-00128-t005:** Summary of research questions, hypotheses, and results.

Research Questions	Hypotheses	Results
RQ1: Which video production style engages the students more emotionally and cognitively?	H1A: Emotional engagement will be higher for the infographic video	Not supported
	H1B: Cognitive engagement will be higher for the infographic video	Supported
RQ2: Is there a difference in learning between conditions?	H2: Learning performance will be higher for the infographic video	Supported
RQ3: What is the relationship between engagement and learning outcomes?	H3A: The higher the emotional engagement, the better the student performance	Supported
	H3B: There is a quadratic relationship between cognitive engagement and performance	Supported

## Data Availability

The data presented in this study are available on request from the corresponding author.

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
