# Peer review of "The Influence of Video Format on Engagement and Performance in Online Learning"

_brainsci, 2021, doi:10.3390/brainsci11020128_

Round 1
Reviewer 1 Report
In general, this is a well-written paper on an interesting experiment, with an extensive literature review. However, there are a number of issues that need to be addressed:
First, although the title of the paper speaks about online learning in general, the text itself speaks almost exclusively of MOOCs, and the vast majority of online learning, at least in North American universities, is not done through MOOCs. My feeling is that the authors can simply downplay the relevance of MOOCs in the text, as most online learning, even in North American universities, does involve some form of video. This of course means that the issue of retention becomes less important but then one can argue that the problem with the lack of retention in MOOCs may be due to other factors, such as the fact that completion of a MOOC does not bring with it the same level of reward in terms of potential credentials as completion of an online course in a more traditional degree program.
Second, the videos that are compared in the study are relatively short (14 minutes). While this is probably the maximum length that should be used for videos in an online environment (this is of course another research question), in many cases, the videos in online courses are much longer in length. They certainly are in MOOCs. It would be interesting to see whether the results hold for longer videos. I would expect them to do but this is an empirical question.
Third, the video capture for the lecture condition may not reflect the lecture videos that are most typically included in MOOCs. It is for example not clear whether the videos capture any of the audio-visual aids that one would typically use in a lecture, such as PowerPoint slides, or even notes on a white or blackboard.
Finally, the number of participants in the study seems extremely low (24 overall, i.e. 12 for each condition). It would be useful to re-run the experiment with larger groups.
Author Response
In general, this is a well-written paper on an interesting experiment, with an extensive literature review. |
Thank you for your remark. |
However, there are a number of issues that need to be addressed: First, although the title of the paper speaks about online learning in general, the text itself speaks almost exclusively of MOOCs, and the vast majority of online learning, at least in North American universities, is not done through MOOCs. My feeling is that the authors can simply downplay the relevance of MOOCs in the text, as most online learning, even in North American universities, does involve some form of video. This of course means that the issue of retention becomes less important but then one can argue that the problem with the lack of retention in MOOCs may be due to other factors, such as the fact that completion of a MOOC does not bring with it the same level of reward in terms of potential credentials as completion of an online course in a more traditional degree program. |
Thank you for your comments. We agree, and we have modified the paper (abstract, introduction and conclusion) to make our paper broader in scope and address online learning in general. |
Second, the videos that are compared in the study are relatively short (14 minutes). While this is probably the maximum length that should be used for videos in an online environment (this is of course another research question), in many cases, the videos in online courses are much longer in length. They certainly are in MOOCs. It would be interesting to see whether the results hold for longer videos. I would expect them to do but this is an empirical question.
|
Thank you for this comment. We agree that the maximum length is an interesting but different research question. And it would be interesting to see if our results hold for longer videos. In our case, we based the duration of our experimental stimuli on previous literature [10,24,32]. Precisely, we have followed the following recommendation: "Based on our data from single watching sessions, a rule of thumb for maximum video length would be in the range of 12-20 minutes." [32] (p.16). However, we also further discuss that element in the limitations section.. |
Third, the video capture for the lecture condition may not reflect the lecture videos that are most typically included in MOOCs. It is for example not clear whether the videos capture any of the audio-visual aids that one would typically use in a lecture, such as PowerPoint slides, or even notes on a white or blackboard. |
Thank you for this comment. We choose a lecture capture condition that would represent the case of a simple and inexpensive recording in a classroom (since it would require no editing or specialized software to make it available to the students). However, we agree that a wide range of alternatives exists between our conditions. Future research should investigate if our results hold when audio-visual aids are edited in a lecture capture format. We now discuss this perspective in our conclusion. |
Finally, the number of participants in the study seems extremely low (24 overall, i.e. 12 for each condition). It would be useful to re-run the experiment with larger groups. |
Our sample is typical of experimental research using neuroscientific measures such as EEG. A recent literature review in the field of applied neuroscience, such as educational neurosciences, suggests that our sample size is within the acceptable range (https://dl.acm.org/doi/10.1145/3410977.3410980). Moreover, we accounted for sample size using the appropriate statistical methods, e.g. Signed rank test, Spearman correlation coefficient, explained in section 4.5 |
Reviewer 2 Report
The paper is well written and technically sound, highlighting results and some limitations of the study. The statistical analyses have shown the variation of the influences of the experimental conditions for the samples studied. I am happy with the findings . However, I have a few comments, particularly on the study participants, instruments, and study limitations. I recommend a revision addressing the following: 1. Please, provide more information about the demographic characteristics of the study participants, including gender and age distributions across the samples. This suggestion aims to make the conclusions more affordable to the target groups intended; 2. Provide information about the validity and reliability of the instruments, and how you maintained them in your study. It is important to highlight the validity and reliability of the Self-Assessment Manikin (SAM) you used to capture self-reported arousal and the valence of participants, and learning performance (achievement) - a multiple-choice questionnaire. 3. In the methods and materials section, please describe the lab setting, how the experiment was carried out, maybe integrating with ‘4.2. Experimental stimuli’. This aims at distinguishing the effect of the experiment, relative to the features of the settings in which it occurs; 4. Please, highlight the limitations of the study before the conclusions section. Currently, the study limitations appeared at the very end of the conclusions. Also, highlight which results could be generalized to other settings to emphasize the contributions of the study beyond the specific characteristics of the experiment examined. I hope these suggestions are of use to the author(s).
Author Response
The paper is well written and technically sound, highlighting results and some limitations of the study. |
Thank you for your remark. |
The statistical analyses have shown the variation of the influences of the experimental conditions for the samples studied. I am happy with the findings . |
Thank you. |
However, I have a few comments, particularly on the study participants, instruments, and study limitations. I recommend a revision addressing the following: 1. Please, provide more information about the demographic characteristics of the study participants, including gender and age distributions across the samples. This suggestion aims to make the conclusions more affordable to the target groups intended; |
Thank you for this comment. We have added the age range and gender and age distribution for each condition. |
2. Provide information about the validity and reliability of the instruments, and how you maintained them in your study. It is important to highlight the validity and reliability of the Self-Assessment Manikin (SAM) you used to capture self-reported arousal and the valence of participants, and learning performance (achievement) - a multiple-choice questionnaire. |
Thank you for this question. The Self-Assessment Manikin (SAM) is a widely used instrument to measure self-perceived emotional response. This measure is widely cited (more than 7000 in google scholar). Because the SAM scale is iconographic and contains only a single item, research shows that it captures a more spontaneous emotional perception (https://doi.org/10.1016/j.chb.2012.12.009). We have rephrase 4.3.1 section to clarify the nature of this instrument. The authors developed the learning performance measure following previously established guidelines in educational research [8, 46]. The development of this measure is explained in section 4.3.3. |
3. In the methods and materials section, please describe the lab setting, how the experiment was carried out, maybe integrating with ‘4.2. Experimental stimuli’. This aims at distinguishing the effect of the experiment, relative to the features of the settings in which it occurs; |
Thank you for this observation. We have added a paragraph on the lab setting in section 4.4. |
4. Please, highlight the limitations of the study before the conclusions section. Currently, the study limitations appeared at the very end of the conclusions. Also, highlight which results could be generalized to other settings to emphasize the contributions of the study beyond the specific characteristics of the experiment examined. I hope these suggestions are of use to the author(s). |
We have moved an expanded version of our limitations to section 5.6. |
Reviewer 3 Report
I appreciated the opportunity to review “The influence of video format on engagement and learning in online learning”. The manuscript was well written and well organized, grounded in a sound theoretical framework. The data presentation was clear, and the research questions are timely, given the shift to online learning because of Covid-19. I offer only minor suggestions below to strengthen this already very fine manuscript, which I believe offers an important contribution to education research.
- What is meant by “lively” on page 3?
- Given that Fredricks, Blumenfeld, and Paris recommend studying all three dimensions of engagement simultaneously, why was behavioral engagement not included in this study? The implication seems to be that passive video watching does not involve or induce behavioral engagement? The authors may consider adding a sentence to this effect or adding a justification for their lack of attention to behavioral engagement in this study.
- In a few places, the authors suggest that infographic videos provide more feedback to students (p. 3, 4, 8, 11). How is feedback being defined here?
- I was intrigued by the differences between the EDA, facial recognition, and SAM data. In section 4.3.1, please include a discussion of whether/how the SAM has been validated, some sample items, and the reliability results of the measure in this particular study.
- I was unclear about how overall vs. over time measures were calculated. Please include a sentence or two of explanation in section 4.5.
- On p. 15, the authors write, “When students were aroused, they activated their senses and resources to pay more attention to the teaching content and process information.” This sentence seems to suggest that emotional engagement leads to cognitive engagement, which leads to performance; however, I don’t believe the authors studied the relationship between emotional and cognitive engagement in this study, so this sentence may need to be reconsidered and revised. Also, there is a typo in the last word on this page.
- Given that a quadratic relationship was found between attention and performance, can you either include the model or indicate at what point performance peaks? The phrase “at some point” is used throughout the manuscript, but the results could be more precise, raising implications for optimal video length.
- On p. 14, the authors mention that pre-test results were similar for both groups. More specificity would be appreciated. Please include the t-test results to show there was not a significant difference in pre-test performance overall and for difficult questions.
- There seems to be assumptions made throughout the manuscript that the infographics in rich media are high quality. On page 5, for example, (I believe in reference to Chen’s research,) the authors write: “Objects which appear on the screen such as tables and images are well-organized in the infographic condition.” However, one can imagine a video with graphics that are not well organized or clear. In the present study, the authors deem the infographics used in the experiment as “interesting” (p. 11), and thereby emotionally engaging. I did wonder about the criteria the authors used to determine the appropriateness of their infographics. It is possible that graphics could be stale, juvenile, offensive, or even confusing—and thereby distracting, off-putting, and not conducive to learning. Are these possibilities accounted for in media richness theory? And in section 4.1, could you offer a little more insight into how graphics were chosen for the video and what made them (in your estimation) interesting?
- While the final section in which implications for future research were drawn out was well done, I would advise a paragraph acknowledging some of the other limitations of the present study, including the small sample size and the operationalization of learning as immediate performance on multiple choice items (as opposed to long-term retention or ability to explain core concepts.)
- I encourage the authors to frame their study not just with the rise of MOOCs, but also in light of the move to online learning precipitated by the pandemic. Because of the pandemic, the need to understand how best to support student engagement and learning in online formats has become all the more urgent, across education sectors.
Author Response
I appreciated the opportunity to review “The influence of video format on engagement and learning in online learning”. The manuscript was well written and well organized, grounded in a sound theoretical framework. The data presentation was clear, and the research questions are timely, given the shift to online learning because of Covid-19. I offer only minor suggestions below to strengthen this already very fine manuscript, which I believe offers an important contribution to education research. |
Thank you for these comments. |
What is meant by “lively” on page 3? |
The sentence was reworked and clarified. |
Given that Fredricks, Blumenfeld, and Paris recommend studying all three dimensions of engagement simultaneously, why was behavioral engagement not included in this study? The implication seems to be that passive video watching does not involve or induce behavioral engagement? The authors may consider adding a sentence to this effect or adding a justification for their lack of attention to behavioral engagement in this study. |
Thank you for this comment. We have added the following sentence in section 2.3 ; “Given this passive research's context (i.e. online videos without behavioural interaction with instructors and peers), we have excluded this dimension from our study to focus on emotional and cognitive engagement.” |
In a few places, the authors suggest that infographic videos provide more feedback to students (p. 3, 4, 8, 11). How is feedback being defined here? |
Thank you for this remark. While the first criteria in the media richness theory involved feedback, our study did not involve any feedback. However, our infographic condition provided visual cues (second criteria) and diverse vocabulary (third criteria). We, therefore, theorize hypothesis H1A and B only on those two criteria and not feedback. We have modified section 3 accordingly. |
I was intrigued by the differences between the EDA, facial recognition, and SAM data. In section 4.3.1, please include a discussion of whether/how the SAM has been validated, some sample items, and the reliability results of the measure in this particular study. |
Thank you for this comment. See response to reviewer 2. The Self-Assessment Manikin (SAM) is a widely used instrument to measure self-perceived emotional response. This measure is widely cited (more than 7000 in google scholar). Because the SAM scale is iconographic (non verbal, pictorial assessment) and contains only a single semantic item, therefore, it is not possible to report reliability as we can with a multi-item psychometric scale. However, the sensibility of the scale has been validated by multiple authors, including those who initially proposed this scale, Lang et al. (doi:10.1016/0005-7916(94)90063-9), using affective picture databases. |
I was unclear about how overall vs. over time measures were calculated. Please include a sentence or two of explanation in section 4.5. |
Thank you for this observation. We have added more details on the calculation. |
On p. 15, the authors write, “When students were aroused, they activated their senses and resources to pay more attention to the teaching content and process information.” This sentence seems to suggest that emotional engagement leads to cognitive engagement, which leads to performance; however, I don’t believe the authors studied the relationship between emotional and cognitive engagement in this study, so this sentence may need to be reconsidered and revised. Also, there is a typo in the last word on this page. |
We agree. We removed this speculative sentence. |
Given that a quadratic relationship was found between attention and performance, can you either include the model or indicate at what point performance peaks? The phrase “at some point” is used throughout the manuscript, but the results could be more precise, raising implications for optimal video length. |
Thank you for this comment. We have provided, in section 5.5, more precise information on the inflection point of attention and when the performance peaks. |
On p. 14, the authors mention that pre-test results were similar for both groups. More specificity would be appreciated. Please include the t-test results to show there was not a significant difference in pre-test performance overall and for difficult questions. |
Thank you for this observation. We have added information about the pre-test comparisons in section 5.3. |
There seems to be assumptions made throughout the manuscript that the infographics in rich media are high quality. On page 5, for example, (I believe in reference to Chen’s research,) the authors write: “Objects which appear on the screen such as tables and images are well-organized in the infographic condition.” However, one can imagine a video with graphics that are not well organized or clear. In the present study, the authors deem the infographics used in the experiment as “interesting” (p. 11), and thereby emotionally engaging. I did wonder about the criteria the authors used to determine the appropriateness of their infographics. It is possible that graphics could be stale, juvenile, offensive, or even confusing—and thereby distracting, off-putting, and not conducive to learning. Are these possibilities accounted for in media richness theory? And in section 4.1, could you offer a little more insight into how graphics were chosen for the video and what made them (in your estimation) interesting? |
Thank you for those observations. Images for the infographic video were chosen based on the guidelines provided by edX Building and Running an edX Course (https://edx.readthedocs.io/). This is now indicated in section 4.2. We understand the concern about the quality of the rich media. We have not manipulated this factor, but it would certainly be an interesting research question. We have added this in the “future research” section. |
While the final section in which implications for future research were drawn out was well done, I would advise a paragraph acknowledging some of the other limitations of the present study, including the small sample size and the operationalization of learning as immediate performance on multiple choice items (as opposed to long-term retention or ability to explain core concepts.) |
Thank you for these comments. We have added these acknowledgements in section 5.5. on the limitation of this experiment. |
I encourage the authors to frame their study not just with the rise of MOOCs, but also in light of the move to online learning precipitated by the pandemic. Because of the pandemic, the need to understand how best to support student engagement and learning in online formats has become all the more urgent, across education sectors. |
We agree and thank you for this comment. We have modified the abstract, the introduction and the conclusion to make our paper more broad and address online learning in general. |
Round 2
Reviewer 1 Report
Thank you for the improvements to the paper.